# Effect of antenatal depression on adverse birth outcomes in Gondar town, Ethiopia: A community-based cohort study

Abel Fekadu Dadi [1,2]*, Emma R. Miller [2], Richard J. Woodman [3], Telake Azale [4], Lillian Mwanri [2]

1 Department of Epidemiology and Biostatistics, Institute of Public Health, College of Medicine and Health Sciences, University of Gondar, Gondar, Ethiopia, 2 Flinders University, College of Medicine and Public Health, Bedford Park, SA, Australia, 3 Center for Epidemiology and Biostatistics, Flinders University, College of Medicine and Public Health, Bedford Park, SA, Australia, 4 Department of Health Promotion and Behavioral Sciences, Institute of Public Health, College of Medicine and Health Sciences, University of Gondar, Gondar, Ethiopia

* Fekten@yahoo.com

## Abstract

### Background

The impact of antenatal depression on pregnancy outcomes has been well investigated in developed countries, but few studies have been conducted in low-income countries. As depression is significantly affected by socio-economic and cultural factors, it would be difficult to generalize evidence from high-income countries to low-income countries. We conducted a community-based cohort study to estimate the incidence of adverse birth outcomes and the direct and indirect pathways via which depression and other psychosocial risk factors may impact such birth outcomes within Gondar town, Ethiopia.

### Methods

The study followed 916 pregnant women who were screened for antenatal depression using the Edinburgh Postnatal Depression Scale (EPDS). We also assessed the incidence of preterm births, Low Birth Weight (LBW) and stillbirths. Modified Poisson regression was used to estimate the relative risk of predictors on adverse birth outcomes and a Generalized Structural Equation Model (GSEM) was used to estimate the direct and indirect effect of antenatal depression and other psychological risk factors on adverse birth outcomes.

### Results

The cumulative incidence of stillbirth, LBW and preterm was 1.90%, 5.25%, and 16.42%, respectively. The risk of preterm birth was 1.61, 1.46, 1.49, and 1.77 times higher among participants who identified as Muslim, reported being fearful of delivery, were government employee's, and who had no antenatal care services, respectively. Partner support moderated the association between depression, preterm birth, and LBW. Depression had no direct effect on birth outcomes but indirectly affected preterm birth via partner support. Religion had both direct and indirect effects on preterm birth, while occupation and fear of delivery

**Data Availability Statement:** All relevant data are within the manuscript.

**Funding:** This research received no specific grant from any funding agency in the public, commercial or not-for-profit sectors.

**Competing interests:** The authors have declared that no competing interests exist.

**Abbreviations:** EPDS, Edinburgh Postnatal Depression Scale; RR, Relative Risk; WHO, World Health Organization; LBW, Low birth weight; CMD, Common Mental Disorder; ODK, Open Data Collection Kit; LNMP, last normal menstrual period; ANC, Antenatal care; OSSS, Oslo Social Support Scale; MUAC, Middle-Upper Arm Circumference; PCI-4, Perinatal Coping Inventory; GLM, Generalised Linear Model; GSEM, Generalized Structural Equation Modeling; AIC, Akaike Information Criteria; BIC, Bayesian Information Criterion; SD, Standard deviation.

had direct effects. The risk of LBW was 9.44 and 2.19 times higher among preterm births and those who had exposure to tobacco, respectively. Stress coping was indirectly associated, and preterm birth and tobacco exposure were directly associated with LBW. The risk of stillbirth was 3.22 times higher in women with antenatal depression and 73% lower in women with higher coping abilities.

## Conclusions

There was a high incidence of all adverse birth outcomes in Gondar Town. Depression and psychosocial risk factors had important indirect negative effects on risk, while partner support provided a positive indirect effect on the incidence of adverse birth outcomes. Interventions that focus on increasing partner engagement and participation in antenatal support may help reduce adverse birth outcomes by enhancing maternal resilience.

## Background

Women with depression during pregnancy produce a higher level of cortisol hormone that can affect fetal growth and subsequently lead to adverse birth outcomes including preterm birth, low birth weight (LBW), and stillbirth [1, 2]. 'Preterm' is defined as birth before 37 weeks of gestation, and approximately 15 million preterm births occur worldwide every year [3]. Approximately one million children die due to preterm complications each year and surviving preterm infants are at high risk for developing life-affecting disabilities, such as visual and hearing problems. More than 60% of preterm births occur in Africa and Asia [4]. Each year in Ethiopia, 320 thousand babies are born preterm, and this is associated with 24,000 dying below the age of five, with a reported preterm prevalence of 4.4% in Gondar, Ethiopia [5, 6].

LBW (birth weight less than 2500gm) is the main cause of perinatal mortality and morbidity [7] and also increases the risk of non-communicable diseases such as diabetes and cardiovascular disease in later life [8]. Globally, more than 15% of all births are low weight with a higher prevalence observed in low-income countries [9]. In Ethiopia, the prevalence of LBW has been estimated at 17.1% in Gondar [10], 14.6% in Tigray [11], and 10% in Adwa General Hospital [12]. In addition to antenatal depression, other risk factors for LBW include maternal nutrition, short maternal stature, extreme age, parity, and short birth space [10, 11].

Antenatal depression leads to the production of a high level of cortisol, a stress hormone, associated with discontinuation of the proper functioning of the placenta and reduced oxygen supply to the fetus which can ultimately result in fetal death or miscarriage [13]. Antenatal depression can have both direct and indirect effects on stillbirth stemming from infection from syphilis, non-communicable diseases, malnutrition, and lifestyles that promote unhealthy pregnancy [14, 15]. In 2015, 2.6 million stillbirths were reported, mostly in low and middle-income countries, three-quarters of which were in Sub-Saharan Africa and Asia [16]. Stillbirth represents a substantial burden in terms of the psychosocial cost on families and the socioeconomic cost for a country [17]. As a country, Ethiopia has the fifth-highest annual number of stillbirths (97,000) globally [18].

A review of studies conducted in developed countries reported maternal depression as a risk factor for poor infant growth [19]. Further reviews from middle- and low-income countries indicate that the risk of preterm birth and LBW is higher among pregnant women who

have experienced depression, anxiety and stress during their pregnancy [3, 20–24]. A preventive effect of social support and stress coping ability from adverse birth outcomes was implicated in previous studies [25, 26]. However, their moderating or mediating effect in the pathway between antenatal depression and adverse birth outcomes was not well studied in Ethiopia.

Studies from Ethiopia have previously reported a lack of association between stillbirth and LBW and maternal common mental disorders (CMD) [27–29]; however, none of these studies has investigated the relationship between antenatal depression and preterm birth. Although Wado and colleagues [28] studied the association between antenatal depression and LBW, their inquiry was conducted in predominantly rural areas. The current community-based cohort study was conducted to identify the incidence and predictors of adverse birth outcomes, focusing on the outcomes resulting from exposure to antenatal depression and other psychosocial risk factors during pregnancy in Gondar Town, Ethiopia. We additionally explored the potential moderating and mediating effects of stress coping and other previously identified risk factors and the mechanisms underlying adverse birth outcomes.

## Methods

### Study design and setting

A community-based cohort study was conducted, and pregnant women in their second or third trimester were followed up until one month after birth. Participants recruitment occurred between June 1, 2018, and March 20, 2019, in Gondar Town. Gondar Town is one of the administrative zones of Amhara Regional State that is located in the Northern part of the region at a distance of 747 km away from Addis Ababa (the capital and largest city of Ethiopia). Gondar town has 12 kebeles (the smallest administrative units in the country), and in 2017/2018, there were 6,450 pregnancies [30, 31]. The town has one government-operated referral hospital, more than eight health centres, and more than 15 private medical clinics [32].

### Data collection and the questionnaire

The primary exposure variable (antenatal depression) and all other covariates were assessed during the second or third trimester of pregnancy. The birth outcomes (stillbirth, low birth weight, and preterm birth) were collected during the first month of delivery. A face-to-face interview was conducted at both times of data collection. Data collectors used a structured and pre-tested electronic-based questionnaire aided by an online Open Data Collection Kit (ODK) application tool [33]. The prepared questionnaire was designed on an excel spreadsheet, converted to XLSForm online, and checked for its validity using Enketo (a preview provided by ODK). The validated form was uploaded onto a Lenovo 7 tablet, and during the data collection phase, data was stored on the Google cloud platform. Nine qualified and registered nurses were trained as data collectors and provided with a Lenovo 7 tablet to administer the questionnaire to the participants. Data were subsequently downloaded from the server and securely stored.

### Instruments

The main exposure variable was antenatal depression, which was assessed using the Edinburgh Postnatal Depression Scale (EPDS). Cox and colleagues developed this scale [34], and it has been adapted and validated for use in the Ethiopian context [35]. Women with an EPDS score of 12 and above were considered to have experienced depression [36–38]. In the current study,

the EPDS demonstrated high reliability for the single construct of distress with internal consistency (α) of 0.74.

The primary outcome was the incidence of adverse birth outcomes, which was determined by the occurrence of at least one of the three conditions of preterm birth, Low Birth Weight (LBW), or stillbirth. Birth weight was obtained from medical records or measured at their home (n = 38, 4.3%) within 24 hours of delivery using a digital SECA scale to the nearest 0.1gm. LBW was classified as a birth weight of less than 2500grams [39]. Gestational age was calculated based on the last normal menstrual period (LNMP) or from ultrasonography information obtained from the women during their ANC service. Preterm birth was considered as birth occurring before 37 completed weeks of gestation [39]. Stillbirth was considered as the death of the fetus after 20 completed weeks of gestation or intrauterine death of a fetus before the onset of labor, or intrauterine death of the fetus during labor and delivery [40].

The Oslo Social Support Scale (OSSS-3) [41] was used to measure maternal social support during pregnancy. OSSS-3 has three items measured by Likert scales, which are summed to 14 points and categorized as 'poor' if the total score is less than nine, and moderate to strong support if the score is 9–14. In this study, OSSS-3 demonstrated high reliability with internal consistency (α) of 0.76.

Partner support was assessed by the statement "My husband helps me a lot" with five possible responses, "Always", "Most of the time", "Some of the time", "Rarely", and "Never". The maternal Middle-Upper Arm Circumference (MUAC) tape was used to measure the nutritional status of the women. The MUAC is a validated and recommended measurement for nutritional status during pregnancy, with a cutoff score between 18 and 22cm as underweight and 22.5 to 31cm as normal [42].

Pregnant women were asked if they participated in moderate-intensity physical activity such as brisk walking, dancing, gardening, and usual housework for two to three hours/week [43] and their responses recorded as yes or no. Exposure to tobacco was assessed by the questions "Have you been smoking since your pregnancy or has there been anybody who smokes near you in your home or your workplace?" If the answer was "yes" for at least one of the questions, the mother was considered to have had a tobacco exposure during pregnancy [44]. Maternal stress coping level was assessed by four customized internally-consistent coping subscales of the Perinatal Coping Inventory (PCI-4), which was specifically developed for pregnancy [45]. In this study, PCI-4 demonstrated moderate reliability with internal consistency (α) of 0.5.

## Sample size

This analysis forms part of a larger mother-child health cohort study designed to explore the extent of perinatal depression and its effect on birth and infant health outcomes in Gondar Town. The required sample size for the whole cohort was calculated using an Epi Info version 7 [46]. This calculation used a two-population proportion formula with the following assumptions: a 95% confidence level, 90% power, exposed to a non-exposed ratio of 1:2, the prevalence of underweight among those free from common mental disorder of 25%, and an effect size of 1.5. A sample size of 809 was required and an additional 20% for expected losses to follow up increased the final sample size to 970.

## Data analysis

As the occurrence of adverse birth outcomes was rare, it was necessary to estimate risk using an appropriate model for rare outcomes. Therefore, we estimated relative risk using a Generalised Linear Model (GLM) with an identity log and binomial link function (log-binomial) [47].

However, similar to other reports [43, 44], we encountered a convergence issue using this approach and therefore changed our model to modified Poisson regression [48]. The advantages of using the modified Poisson regression model include improved precision for estimation of relative risk, robustness to omitted covariates [49, 50], and efficiency when using clustered data [51]. Covariates that were significant at p<0.2 in univariate analysis and those known to be associated with adverse outcomes a priori were included in a multivariate model. We also tested for any potential moderating effects using interaction terms [52]. Both crude and adjusted relative risk estimates, together with 95% confidence intervals were presented for predictor variables with p<0.05.

A mediation analysis using Generalized Structural Equation Modeling (GSEM) was conducted to look at the direct and indirect relationships between the identified covariates and each of the birth outcomes. A GSEM with logit link function was used to allow estimation with the categorical outcome and mediator variables [53, 54]. Relative measures of model fit including the Akaike Information Criteria (AIC) and Bayesian Information Criterion (BIC) were used to refine the initial model that was based on expert knowledge of the causal pathways and the model with the best fit was retained. To obtain the direct, indirect and total effects of the predictors and mediators, we used the user-written ldecomp command. Analyses were performed using Stata version 14.0 (StataCorp, USA) [55]. The direct, indirect, and total effects are reported as unstandardized beta coefficients together with standard errors and p-values [56, 57].

### Ethical approval

Ethical clearance was obtained from the Social and Behavioral Research Ethics Committee (SBREC) of the Flinders University [58] and the Institutional Review Board of the University of Gondar. A support letter was obtained from Gondar town mayoral office and respective kebeles administration offices. Participants of the study were informed about the purpose, objectives and their right to participate, decline participation or withdraw their participation in the research activities and written consent was obtained. Privacy and confidentiality were maintained throughout the study. The participants were informed that they have the right to withdraw from the study at any time. Women who were found to be seriously ill and fulfilled the following criteria were referred to University of Gondar Specialized Hospital Psychiatry unit for further diagnosis and treatment: have an overall scale score of 13 (those with $\geq$ 17 were excluded from the study for further follow up) and those who had a score 1, 2, 3 on item tenth (question about the thought of self-harm) on Edinburgh Postnatal Depression Scale (EPDS) [59].

## Results

During the survey, 960 pregnant women were contacted in the six selected districts, and 44 were excluded because of refusal, absenteeism, and ethical issues. Finally, a total of 916 participants were followed to the postnatal period. During the follow up there were four women who refused to participate, eight were lost to follow up, and nine were excluded because of their high risk of severe depression (defined as an Edinburgh Postnatal Depression Scale (EPDS) score of $\geq$17). Finally, 895 participants were included in the analysis. The survey response rate was 97.7%, and the loss-to-follow up rate was 2.3%.

### Socio-demographic characteristics of the study participants

Table 1 shows the socio-demographic characteristics of the study participants. The mean (SD) age of the participants was 26.5 (4.5) years, and no age difference was found between those

**Table 1. Socio-demographic characteristics of pregnant women included in the study (N = 895), Gondar town, Ethiopia, 2018.**

| Variable/category | Adverse birth outcome | | | p-value |
|---|---|---|---|---|
| | Yes (n = 163), n (%) | No (n = 732), n (%) | Total n = 895, n (%) | |
| Women age at enrolment | | | | 0.84 |
| 18–24 | 51 (31.3) | 231 (31.56) | 282 (31.5) | |
| 25–34 | 99 (60.7) | 452 (61.75) | 551 (61.6) | |
| > = 35 | 13 (8.0) | 49 (6.7) | 62 (6.9) | |
| Mean(±SD) | 26.8 (4.5) | 26.5 (4.5) | 26.5 (4.5) | |
| Household monthly income | | | | 0.76 |
| Low | 76 (17.4) | 361 (82.6) | 437 (48.8) | |
| Medium | 68 (18.6) | 297 (81.4) | 365 (40.8) | |
| High | 19 (20.4) | 74 (79.8) | 93 (10.4) | |
| Monthly income Mean(±SD) | 3650.9(2962.2) | 3485.9(2990.1) | 3516(2984) | |
| Women education | | | | 0.70 |
| None | 17 (10.4) | 97 (13.2) | 114 (12.7) | |
| Primary | 43 (26.4) | 184 (25.1) | 227 (25.4) | |
| High school | 66 (40.4) | 273 (37.3) | 339 (37.9) | |
| Tertiary | 37 (22.7) | 178 (24.3) | 215 (24.0) | |
| Women occupation | | | | 0.23 |
| Home duties | 107 (65.6) | 532 (72.7) | 639 (71.4) | |
| Student | 2 (1.2) | 13 (1.8) | 15 (1.7) | |
| Government employee | 27 (16.6) | 100 (13.7) | 127 (14.2) | |
| Self-employee | 27 (16.6) | 87 (11.9) | 114 (12.7) | |
| Women religion | | | | 0.01 |
| Orthodox Christian | 117 (71.8) | 597 (81.6) | 714 (79.8) | |
| Muslim | 45 (27.6) | 128 (17.5) | 173 (19.3) | |
| Protestant Christian | 1 (0.6) | 7 (0.96) | 8 (0.89) | |
| Women marital status | | | | 0.43 |
| Single | 3 (1.8) | 16 (2.2) | 19 (2.1) | |
| Partnered | 159 (97.5) | 701 (95.8) | 860 (96.1) | |
| Separated | 1 (0.61) | 15 (2.0) | 16 (1.8) | |
| Difficulty accessing food in the last three months | | | | 0..91 |
| Yes | 7 (4.3) | 30 (4.1) | 37 (4.1) | |
| No | 156 (95.7) | 702 (95.9) | 858 (95.9) | |

p-value was calculated based on chi-square test statistics

with and without adverse birth outcomes (p = 0.48). There were 714 (79.8%) who described their religion as Orthodox Christian and 860 (96.1%) who were married. There were 339 (37.9%) participants who had secondary education and 639 (71.4%) who occupied with unpaid home duties. The mean (SD) monthly income of the participants was 3516 (2984) Ethiopian birr (140 AUD). There was no age (p = 0.48) or income (p = 0.52) differences between women with and without adverse birth outcomes. There was, however, a significant difference in religious categories between those with and without an adverse birth outcome (p = 0.012).

## Obstetric and behavioral characteristics of the study participants

Table 2 shows the obstetric and behavioral characteristics of the study participants. Overall, 763 (85.2%) of the pregnancies were planned, 341 (38.1%) were first pregnancies, and 509 (91.9%) had no history of adverse birth outcomes. There were 857 (95.7%) participants who

**Table 2. Obstetric and behavioral characteristics of study participants included in Gondar town, Ethiopia, 2018.**

| Variable/category | Adverse birth outcome | | Total n = 895, n (%) | p-value |
|---|---|---|---|---|
| | Yes (n = 163), n (%) | No (n = 732), n (%) | | |
| Pregnancy intention | | | | 0.57 |
| Planned | 141 (86.5) | 622 (85.0) | 763 (85.2) | |
| Unplanned | 22 (13.5) | 110 (15.0) | 132 (14.8) | |
| Parity of the mother | | | | 0.68 |
| 1 | 58 (35.6) | 283 (38.7) | 341 (38.1) | |
| 2 | 56 (34.4) | 228 (31.1) | 284 (31.7) | |
| 3–8 | 49 (30.1) | 221 (30.2) | 270 (30.2) | |
| Mean(±SD) | 2.1 (1.1) | 2.1 (1.2) | 2.1 (1.2) | |
| Previous adverse birth history | | | | 0.07 |
| Yes | 13 (12.4) | 32 (7.1) | 45 (8.1) | |
| No | 92 (87.6) | 417 (92.9) | 509 (91.9) | |
| Antenatal care service uptake (at least one) | | | | 0.18 |
| Yes | 153 (93.9) | 704 (96.2) | 857 (95.7) | |
| No | 10 (6.1) | 28 (3.8) | 38 (4.3) | |
| Fearful about the delivery | | | | 0.02 |
| Yes | 46 (28.2) | 145 (19.8) | 191 (21.3) | |
| No | 117 (71.8) | 587 (80.2) | 704 (78.7) | |
| Undertook physical activity | | | | 0.86 |
| Yes | 160 (98.2) | 717 (97.9) | 877 (98.0) | |
| No | 3 (1.8) | 15 (2.0) | 18 (2.0) | |
| Exposure to tobacco | | | | 0.89 |
| Yes | 17 (10.4) | 79 (10.8) | 96 (10.7) | |
| No | 146 (89.6) | 653 (89.2) | 799 (89.3) | |
| Exposure to coffee | | | | 0.50 |
| Daily | 66 (40.5) | 303 (41.4) | 369 (41.2) | |
| Sometimes | 62 (38.0) | 247 (33.7) | 309 (34.5) | |
| Never | 35 (21.5) | 182 (24.9) | 217 (24.3) | |
| Nutritional status of the mother | | | | |
| Underweight (MUAC 18–22) | 20 (12.3) | 109 (14.9) | 129 (14.4) | |
| Normal (MUAC 22.5–31) | 143 (87.7) | 623 (85.1) | 766 (85.6) | |
| MUAC (Mean(±SD) | 24.4 (1.6) | 23.9 (1.7) | | ≤0.01 |

p-value was calculated based on chi-square test statistics

attended Antenatal Care (ANC) service, 704 (78.7) who did not have a fear about the delivery, and 766(85.6%) who were underweight. Almost all participants, 877 (98.0%) undertook physical activity during pregnancy. Seven hundred ninety-nine (89.3%) of the participants were not exposed to tobacco smoke, and 369 (41.2%) consumed coffee daily during pregnancy. Compared to other participants, those who ultimately experienced adverse birth outcomes were more likely to report being fearful about the delivery (p = 0.018) and had higher mean MUAC measurements (p = 0.002).

## Psycho-social characteristics of study participants

Table 3 shows the psycho-social characteristics of the participants. There were 833 (93.1%) participants who had no history of common mental health disorders, 716 (80.0%) who reported good social support, and 842 (94.1%) who reported receiving good support from

**Table 3. Psycho-social characteristics of pregnant women included in the study (N = 895), Gondar town, Ethiopia, 2018.**

| Variable/category | Adverse birth outcomes | | Total n = 895, n (%) | p-value |
|---|---|---|---|---|
| | Yes (n = 163), n (%) | No (n = 732), n (%) | | |
| History of common mental disorder | | | | 0.43 |
| Yes | 9 (5.5) | 53 (7.2) | 62 (6.9) | |
| No | 154 (94.5) | 679 (92.8) | 833 (93.1) | |
| Social support | | | | 0.44 |
| Good | 134 (82.2) | 582 (79.5) | 716 (80.0) | |
| Poor | 29 (17.8) | 150 (20.5) | 179 (20.0) | |
| Social support scale (Median(±IQR)) | 11 (9–13) | 11 (9–13) | | |
| Internal consistency (α) | 0.76 (high reliability) | | | |
| Partner support | | | | ≤0.01 |
| Always | 95 (58.3) | 319 (43.6) | 414 (46.3) | |
| Most of the time | 32 (19.6) | 230 (31.4) | 262 (29.3) | |
| Some of the time | 27 (16.6) | 139 (19.0) | 166 (18.5) | |
| Rarely | 9 (5.5) | 44 (6.0) | 53 (5.9) | |
| Stress coping ability | | | | 0.049 |
| Poor | 113 (69.3) | 447 (61.1) | 560 (62.3) | |
| Good | 50 (30.7) | 285 (38.9) | 335 (37.4) | |
| Stress coping scale, Median (IQR) | 7 (7–9) | 8 (7–18) | | |
| Internal consistency (α) | 0.5 (moderate reliability) | | | |
| Antenatal depression | | | | 0.81 |
| Yes | 13 (8.0) | 45 (6.1) | 58 (6.5) | |
| No | 150 (92.0) | 687 (93.5) | 837 (93.5) | |
| Depression scale (Median(±IQR)) | 4 (1–7) | 4 (2–7) | | |
| Internal consistency (α) | 0.74 (High reliability) | | | |

p-value was calculated based on chi-square test statistics

their partner. More than half of the participants, 560 (62.3%), had poor stress coping ability, and 58 (6.5%) had moderate depression symptoms. Participants experiencing adverse birth outcomes more often reported poor partner support (p = 0.005) and reduced stress coping ability (p = 0.049). However, the percentage of those with minor depression was not different between those with and without adverse birth outcomes.

## Incidence proportion of adverse birth outcomes

From the 895 pregnancies, 17 resulted in stillbirth (1.90%; 95%CI: 1.11–3.02), 47 in LBW (5.25%; 95%CI: 3.88–6.92), and 147 resulted in preterm births (16.42%; 95%CI: 14.05–19.01). There was a significant difference in the proportion of preterm births across the six districts (p = 0.001). The highest proportion of preterm birth (55.7%) was in the Abiygzi district and the lowest (4.0%) was in the Kirkos district. The proportion of LBW and stillbirths was not statistically different in selected districts (p = 0.951 and p = 0.538, respectively).

## Predictors of preterm birth

Table 4 shows the results of multivariate analysis for pre-term birth. In the fully adjusted model of pre-term birth, religion, occupation, antenatal care service, stress coping ability, partner support, and fear of delivery were significant independent predictors of preterm birth. However, there was no difference in the risk of preterm birth between participants with and

**Table 4. Multivariable analysis of adverse birth outcome predictors in Gondar town, 2018 (n = 895).**

| Variables | Preterm birth | | Low Birth Weight | | Still Birth | |
|---|---|---|---|---|---|---|
| | CRR, 95%CI | ARR,95%CI | CRR, 95%CI | ARR,95%CI | CRR, 95%CI | ARR,95%CI |
| Women's religion | | | | | | |
| Orthodox | Reference | | | | | |
| Muslim | 1.61 (1.17, 2.22) | 1.61 (1.16,2.24)* | | | | |
| Women's occupation | | | | | | |
| House wife | Reference | | | | | |
| Government employee | 1.32 (0.91, 1.94) | 1.49 (1.00,2.19)* | | | | |
| Self-employee | 1.42 (0.95, 2.11) | | | | | |
| Had fear of delivery | | | | | | |
| Yes | 1.52 (1.11, 2.09) | 1.46 (1.06,2.01)* | | | | |
| No | Reference | | | | | |
| Antenatal care service uptake | | | | | | |
| Yes | Reference | | | | | |
| No | 1.65 (0.94, 2.86) | 1.77 (1.03,3.03)* | | | | |
| Preterm birth | | | | | | |
| Yes | | | 9.86 (5.53,17.56) | 9.44 (5.06,17.63)* | | |
| No | | | Reference | | | |
| MUAC measurement of the mother | 1.15 (1.06, 1.24) | | | | | |
| Exposure to cigarette smoking | | | | | | |
| Yes | | | 1.97 (0.98,3.95) | 2.19 (1.10, 4.37)* | | |
| No | | | Reference | | | |
| Symptom of Depression | | | | | | |
| No depression | Reference | | | | | |
| Depressed | 1.17 (0.67,2.03) | | 1.72 (0.71,4.18) | | 3.09 (0.91,10.46) | 3.22 (1.04,9.98)* |
| Try to cope stress | | | | | | |
| Poor | Reference | | | | | |
| Good | 0.67 (0.48,0.93) | | | | 0.36 (0.10,1.24) | 0.27 (0.07,0.99)* |

The model has been adjusted for age, income, education, marital status, pregnancy intention, previous adverse history, parity, partner and social support.

* significant at p-value <0.05, CRR: Crude relative risk, ARR: Adjusted relative risk; 95%CI: 95% confidence interval

without depression during pregnancy. Based on the previous study in Ethiopia [28], we also assessed interactions between depression and stress coping and partner support. There was a strong moderating effect of husband support and stress coping ability on preterm birth. Due to very small observations in some categories, some interaction terms for the various categories of partner support X stress coping were not estimable and were therefore omitted. Table 5 presents the results of a moderating effect between partner support and antenatal depression. Accordingly, the risk of having preterm birth was 4.38 (95%CI: 1.25, 15.33) times higher among participants with depression and moderate support and 4.99 (95%CI: 1.28, 19.42)

**Table 5. Relative risk of preterm birth and low birth weight moderated by partner support, in Gondar town, 2018 (N = 895).**

| Variables | Preterm birth | | LBW | |
|---|---|---|---|---|
| | Crude RR 95%CI | Adjusted RR 95%CI | Crude RR 95%CI | Adjusted RR 95%CI |
| Partner support | | | | |
| No depression and good support Depressed and good support | Reference 1.29(0.69,2.39) | 4.38(1.25,15.33)* | Reference not estimable | not estimable |
| Depressed and poor support | 0.58(0.09,3.70) | 4.99(1.28,19.42)* | 2.54(0.39,16.32) | 3.40(1.26,9.17)* |

**Table 6. Direct, indirect, and total effect of antenatal depression on adverse birth outcomes in Gondar town, 2018 (N = 895).**

| Covariates | Direct | | Indirect | | Total effect | |
|---|---|---|---|---|---|---|
| | β(SE) | p-value | β(SE) | p-value | β(SE) | p-value |
| **Preterm birth** | | | | | | |
| Antenatal depression | 0.251(0.126) | 0.571 | | | 0.323(0.40) | 0.420 |
| Indirect (partner support) | | | -0.137(0.051) | 0.007 | | |
| Indirect (fear of delivery) | | | 0.213(0.109) | 0.050 | | |
| Fear of delivery | 0.535(0.216) | 0.013 | | | | |
| Partner support | -0.435(0.169) | 0.01 | | | | |
| Religion | 0.465(0.198) | 0.007 | 0.125(0.052) | 0.017 | 0.590(0.218) | 0.007 |
| Occupation | 0.261(0.126) | 0.038 | | | | |
| Stress coping ability | 0.406(0.208) | 0.051 | | | | |
| **Low birth weight** | | | | | | |
| Antenatal depression | 0.341(0.341) | 0.460 | 0.393(0.308) | 0.202 | 0.735(0.474) | 0.121 |
| Cigarette exposure | 0.961(0.441) | 0.03 | | | | |
| Preterm birth | 2.588(0.339) | 0.001 | | | | |
| Partner support | 0.255(0.329) | 0.439 | -0.234(0.131) | 0.055 | 0.021(0.345) | 0.951 |
| Stress coping | -0.188(0.303) | 0.536 | 0.245(0.110) | 0.026 | 0.057(0.351) | 0.871 |
| **Still Birth** | | | | | | |
| Antenatal depression | 1.265(0.656) | 0.054 | | | | |
| Stress coping | 1.101(0.643) | 0.087 | | | | |

times higher among participants with depression and poor support as compared to participants with no depression and good support. (Table 5).

The results of the mediation analysis are reported in Table 6 and Fig 1. Although there was no direct effect of antenatal depression on preterm birth, depression had a significant indirect effect through partner support (unstandardized β = -0.137, p = 0.007) and mother's fear of delivery (unstandardized β = 0.213, p = 0.05). However, there was no indirect effect of antenatal depression on preterm birth through stress coping. (Table 6 & Fig 1)

In multivariate analysis, the risk of preterm birth was 1.61 (95%CI: 1.16, 2.24) times higher among participants who identified as Muslim. Similarly, in the path analysis, religion had both a direct effect (unstandardized β = 0.465, p = 0.007) and an indirect effect (unstandardized β = 0.125, p = 0.017) on preterm birth with the effect being mediated by maternal stress coping ability and partner support. Participants who expressed fear about the delivery had an increased risk of preterm birth (Adjusted RR = 1.46; 95%CI: 1.06, 2.01) compared to their counterparts. A similar relationship was found in the path analysis where fear of delivery had a direct effect on preterm birth (unstandardized β = 0.535, p = 0.013). (Tables 4 and 6 & Fig 1)

Participants who were government employees had an increased risk of preterm birth (adjusted RR = 1.49; 95%CI: 1.00, 2.19) relative to those who performed home duties only. Similarly, the occupation had a direct positive effect on preterm birth in path analysis (unstandardized β = 0.261, p = 0.038). The risk of preterm birth was 1.77 times higher (95%CI: 1.03, 3.03) in participants who did not have antenatal care visits during pregnancy, although there was no significant relationship in the path analysis. (Tables 4 and 6 & Fig 1)

### Predictors of low birth weight

In multivariate Poisson regression modelling, the risk of having a LBW infant was significantly increased in women with preterm birth and maternal tobacco smoke exposure during pregnancy. The adjusted model is presented in Table 4, and the path analysis in Fig 2, which show

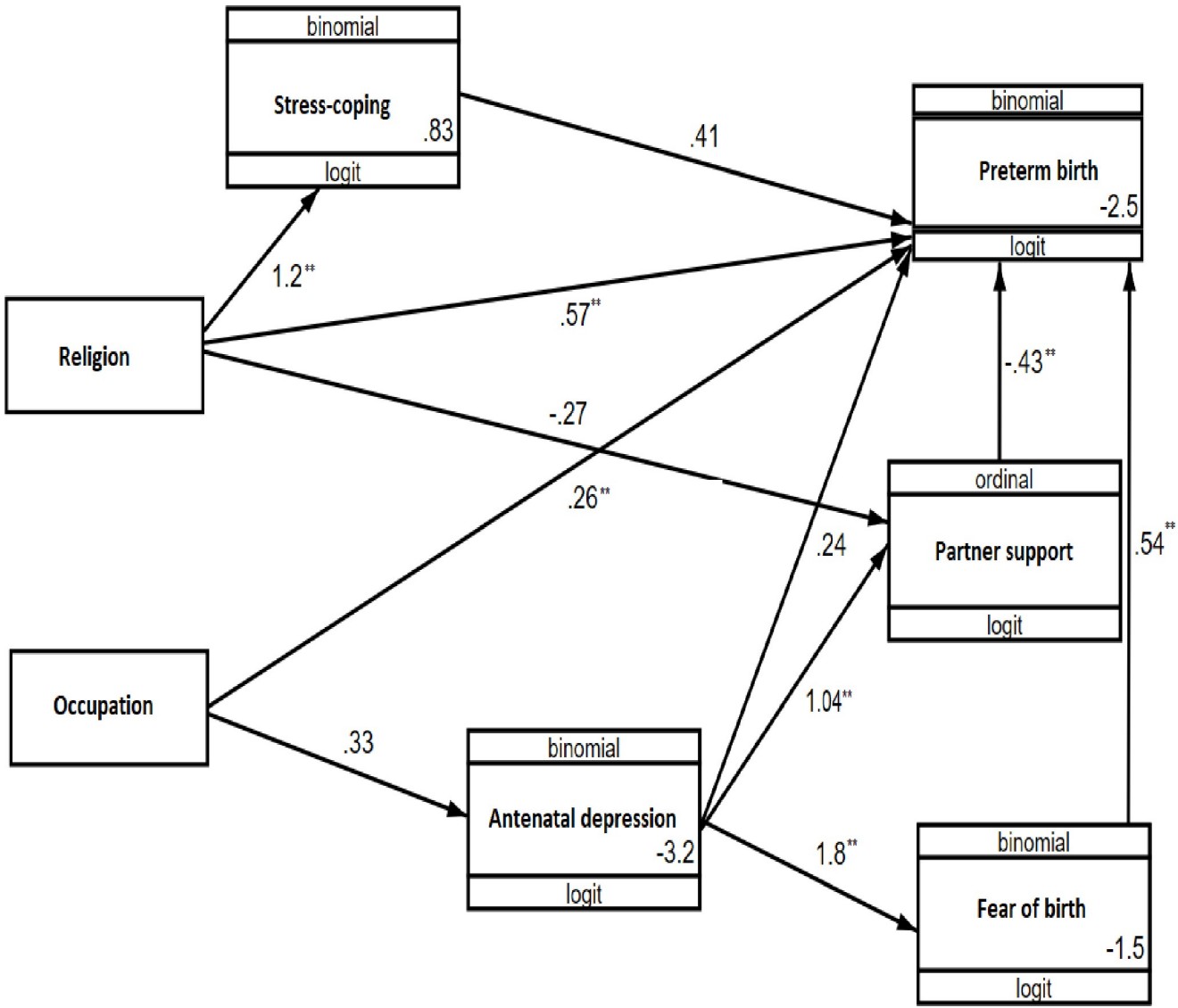

**Fig 1. A path analysis model showing hypothesized causal pathways underlying antenatal depression and preterm birth in Gondar town, 2018 (N = 895).**

the factors affecting LBW as well as their hypothesized causal relationships. Although antenatal depression was not directly associated with LBW, both partner support and stress coping ability moderated the relationship between antenatal depression and the risk of LBW, although the moderating effect was small across all categories of partner support and coping ability. The risk of LBW was 3.4 (95%CI: 1.26, 9.17) times higher among participants with minor depression and poor support relative to participants with no minor depression and good support (Table 5).

There was a significant indirect effect of stress coping ability on LBW (unstandardized β = 0.245, p = 0.026) through preterm birth. Despite significant paths from antenatal depression to LBW through both partner support and tobacco exposure, antenatal depression had no direct (unstandardized β = 0.341, p = 0.460) or indirect effect (unstandardized β = 0.393, p = 0.202) on the risk of LBW. The indirect effects of partner support on LBW through preterm birth and

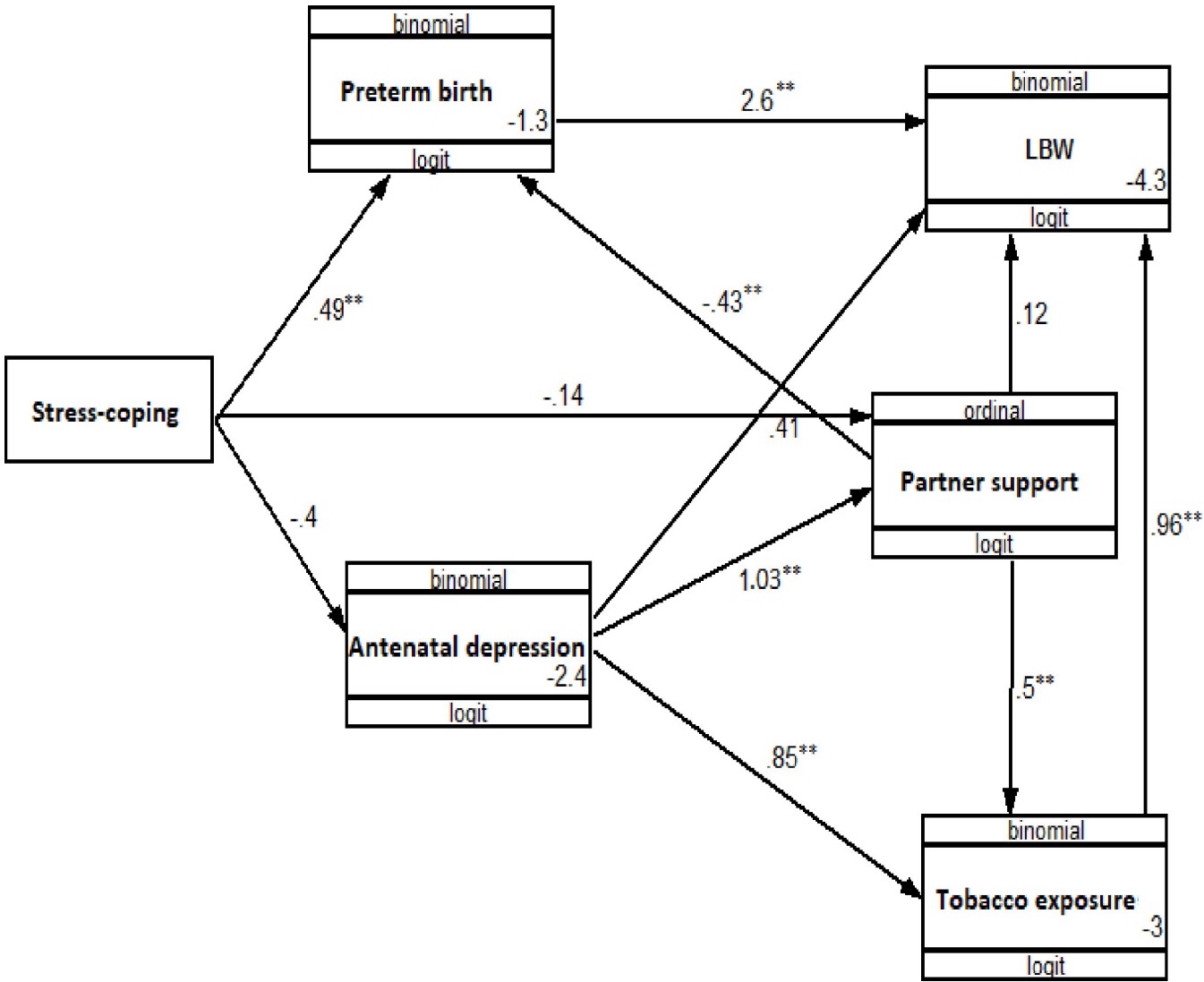

**Fig 2. A path analysis model showing hypothesized causal pathways underlying antenatal depression and low birth weight in Gondar town, 2018 (N = 895).**

tobacco exposure were also not significant (unstandardized β = -0.234, p = 0.055). (Table 6 & Fig 2)

As anticipated, the multivariate-adjusted risk of LBW was 9.44 (95%CI: 5.06,17.6) times higher among preterm births compared to full-term births. This increase in risk was also observed in the path analysis in which preterm birth had a direct positive effect on the risk of LBW (unstandardized β = 2.59, p = 0.001). Preterm birth also mediated an effect of stress coping ability (unstandardized β = 0.245, p = 0.026) and partner support (unstandardized β = -0.234, p = 0.055) on the risk of LBW. In multivariate analysis, women with tobacco exposure during pregnancy had an increased risk of LBW (ARR = 2.19; 95%CI: 1.10, 4.37), an association that was confirmed in the path analysis, in which exposure to tobacco had a direct positive effect on the risk of LBW (unstandardized β = 0.961, p = 0.03). (Tables 4 and 6, & Fig 2)

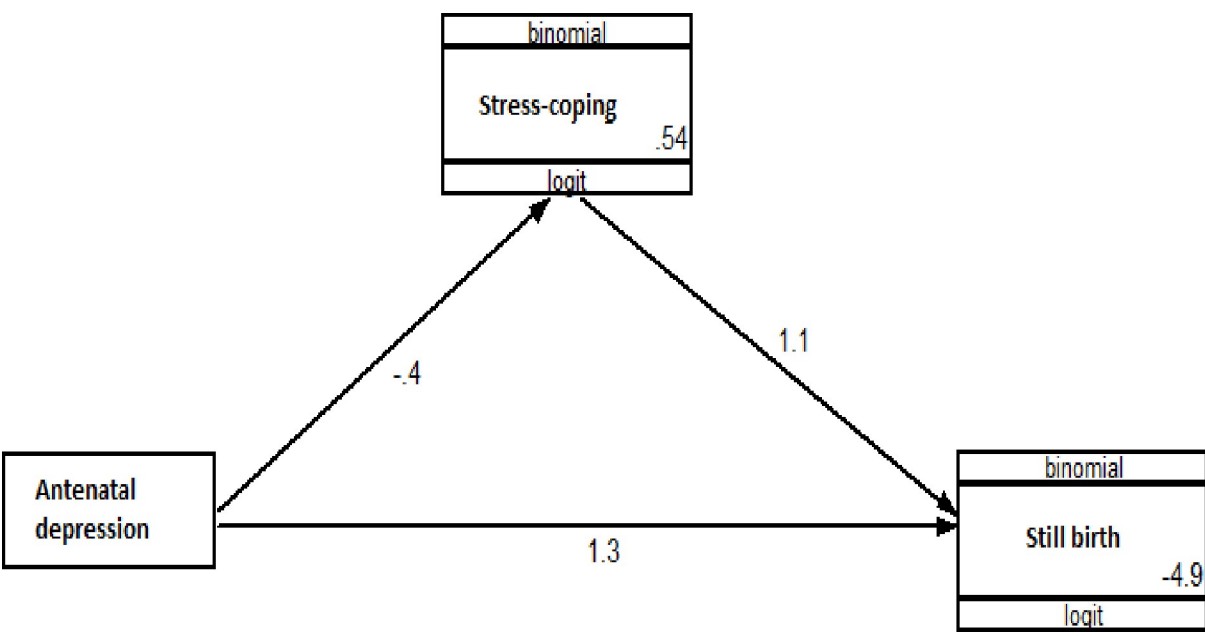

**Fig 3. A path analysis model showing hypothesized causal pathways underlying antenatal depression and stillbirth in Gondar town, 2018 (N = 895).**

### Predictors of stillbirth

The adjusted risk of stillbirth was 3.22 (95%CI: 1.04,9.98) times higher among study participants with depression during pregnancy. The path analysis showed a small direct positive effect of antenatal depression on stillbirth (unstandardized β = 1.265, p = 0.054). However, the risk of stillbirth was decreased by 73% in study participants with good stress coping ability (adjusted RR: 0.27; 95%CI: 0.07, 0.99). However, this effect was also slightly attenuated in the path analysis (unstandardized β = 1.101, p = 0.087). (Tables 4 and 5 & Fig 3)

## Discussion

The short and long-term undesirable effects of maternal depression on the newborn and their women have been well established, and the impact is highest in low-income countries [17, 60]. While previous studies from developed countries have reported the effect of maternal depression during pregnancy on newborn weight, preterm, and stillbirth [16, 61], the results from studies conducted in Ethiopia were inconsistent. As a result, interventions that might improve maternal mental health and, potentially, reduce the incidence of adverse birth outcomes [62] have not yet been implemented in Ethiopia.

In the current community-based cohort study the incidence of stillbirth in Gondar town was 19 per 1000 live births, a finding consistent with a systematic review of studies in Ethiopia [63], Ethiopian Demographic and Health Survey data (EDHS, 2011) [64], and with the national estimate [65]. However, the incidence is lower than that reported in some other studies including the Ethiopian Mini Demographic and Health Survey (EMDHS) for Amhara region [66], a health institution based cohort study conducted in rural Oromia region [67], and an institution based study in Gondar town [68]. The variation in estimated incidence is most likely related to differences in the study design, setting, and period. Preventable stillbirths, premature newborns, and child mortality have been identified as a major area for a high return on investment in the Global Strategy for Women's, Children's and Adolescents'

Health (2016–2030). This strategy set a target to reduce newborn mortality to fewer than 12 per 1000 births and a reduction in stillbirths to fewer than 10 per 1000 births in every country [69]. However, estimates from the current and previous studies show how Ethiopia is still lagging behind other countries in reducing still birth [18]. Possible reasons for the higher number of still births include an under-developed health infrastructure, a lack of trained health professionals, a lack of focus on the problem [70, 71] and, potentially overlooking the issue of maternal mental health in the pregnancy continuum of care.

In this study, the risk of stillbirth was 3.22 times higher in study participants who had depression. Risk of stillbirth was 0.27 time lower among study participants with good stress coping ability. However, in the path analysis, both depression and stress coping ability were not statistically significant, most likely as a result of insufficient power due to a relatively low number of stillbirths. While a lack of association between antenatal depression and stillbirth has been reported in high-income countries [72, 73] as well as in Ethiopia [74], plausible biological mechanisms exist for a causal effect of antenatal depression on stillbirth. Depressed women are more likely to produce a high level of stress hormone (cortisol), which could affect the proper functioning of the placenta, reducing oxygen supply to the fetus and, ultimately resulting in fetal distress or death [13]. Another environmental explanation could be that pregnant women with signs of depression generally have low health-seeking behaviors for pregnancy-related problems or other infections associated with stillbirth [29, 75]. The mother's ability to cope with stress might also lower the risk of still birth with a logical explanation being that a higher level of stress coping ability reduces the risk of developing depression since it buffers the impact of stressors. The uncertainty in our results suggest that further studies with a larger sample size are warranted.

The estimated incidence of LBW in our study was 5.2% (21.0% in preterm and 2.1% in term births), lower than in other community [28] and health institution based studies [67], and also lower than reported in health institutions elsewhere in Ethiopia [68, 76, 77]. However, our study differed from previous studies, being an urban setting with relatively large sample size. The previous studies were also institutionally based, which may bias the true prevalence of LBW within the wider community. Given the multifaceted effects of LBW [78], even our lower estimated incidence still has the potential for large impacts on health at the population level.

Both partner support and a higher stress coping ability modified the association between antenatal depression and LBW, although the effect for stress coping was relatively small. The risk of LBW was 3.4 times higher among study participants with depression and poor partner support relative to those without depression and good partner support. Besides, partner support and preterm birth mediated indirect effects of antenatal depression on LBW. A similar mediating effect for antenatal depression of social support on low birth weight has been reported in other cohort studies [28, 79]. The significant indirect effect of stress coping ability on LBW mediated by preterm birth and antenatal depression has been previously highlighted [24, 80, 81]. However, other reviews report inconsistent associations for antenatal depression and low birth weight [82, 83]. Maternal mental wellbeing might affect intrauterine nutritional conditions which could be subsequently reflected in LBW [84, 85]. Together, these findings suggest that strengthening partner support through enhancing male involvement in maternal care and women counselling might enhance stress coping ability and considerably reduce the risk of depression and LBW.

As expected, the risk of LBW was 9.44 times higher among preterm births (21.1% among preterm births and 2.1% among term births). The association between LBW and preterm birth was also observed in similar studies conducted in Gondar [68] and Dangla hospital [86]. However, a lack of any association between preterm birth and the risk of low birth weight was

reported in Europe [87]. Though the risk factors for preterm birth and LBW are entirely differ-ent, the interventions taken to reduce preterm birth may be equally important to reduce the risk of LBW [88].

The risk of LBW was 2.19 times higher among study participants who had tobacco smoke exposure during pregnancy. Tobacco exposure as a risk factor for low birth weight has been reported in studies from developed countries [87, 89, 90]. Potential mechanisms underlying this association include an increased concentration of cotinine and nicotine in the amniotic fluid of the newborn leading to nicotine-induced placental vasoconstriction, a reduced blood oxygen uptake, increases in carboxyhemoglobin, or an increased occurrence of placental vas-cular disease, all of which affect fetal growth [91]. According to the Ethiopian Demographic and Health Survey (EDHS, 2016), the prevalence of smoking in women was 1% [92], but the prevalence in school-based studies is much higher and the risk of second-hand smoking at home was reported to be 62.5% [93, 94]. Similarly, in the current study, 10.7% of pregnant women had tobacco exposure, suggesting that tobacco is becoming a major public health trait in Ethiopia. Depression might also lead the mother to continue smoking during pregnancy, as suggested by the path analysis. While the detrimental effects of tobacco smoking on overall health have already been well established, adherence to the tobacco control strategy developed by WHO and the government of Ethiopia is important to prevent an increase in LBW children [95].

Within Ethiopia, an estimated 320,000 (10.0%) babies are born prematurely each year [5], similar to that within our study where the incidence was 16.4%. Other studies in various regions of the country have reported similar findings [67, 68, 77, 96–98] with estimates much higher than the global preterm rate of 10.6% [99]. Within our study, the district with the largest proportion of preterm birth (55.7%) was in Abygizi. Relative to other districts, the women in this district generally had reduced stress coping abilities (88%), identified as Muslims (49.5%), were in paid work (35%), were fearful of delivery (32%) and attended no ANC services (5%), all of which were factors identified as increasing the risk of preterm birth. Other factors that might also lead to such variation in premature delivery that were not considered include dis-tance and accessibility to a health facility, infections, or other gynecological and obstetric con-ditions [96, 97].

Preterm birth is a major cause of death and long-term loss of human potential and its com-plication is a direct cause of neonatal death [100]. Preterm neonatal deaths were reported to be as high as 32% in Gondar University Comprehensive Specialized Hospital [101]. Despite a reduction over the last few years, the high incidence of preterm birth persists [5]. In addition to routinely practiced clinical and public health service interventions, psychosocial interven-tions might also reduce the behavioral, mental, and psychological risk factors of preterm birth [102, 103].

Unlike other recent studies [104–106], there was no overall direct association between ante-natal depression and preterm birth in the current study. However, depression was modified by partner support and stress coping ability, and preterm birth was 4.38 and 4.99 times higher, respectively, in those with poor partner support and poor stress coping ability. Similarly, in the path analysis, depression had a significant indirect effect on preterm birth via partner support and fear of delivery, each of which also had separate direct positive effects on preterm birth. In conclusion, partner support buffers the association between depression and the risk of preterm birth, a finding consistent with a review of randomized controlled studies that suggests offer-ing support for at-risk pregnant women [107]. A positive effect of behavioral therapy during pregnancy on good birth outcomes [104] and a buffering effect of perceived support from a partner [79] was also observed in similar cohort studies.

Low and preterm births are associated with restricted fetal growth [16, 70, 103] with plausible biological mechanisms to explain these relationships. Depression during pregnancy stimulates the hypothalamus-pituitary-adrenal (HPA) system leading to an increased cortisol hormone secretion that restricts fetal growth [2, 108–110]. Depression during pregnancy also increases the production of corticotrophin-releasing hormone(CRH) from the placenta, initiating premature labor, a scenario termed as the Placental Clock [111]. Depressed women are also more likely to have inflammation or infection of reproductive organs [112, 113], low heath service uptake, and poor health behavior [114–116] all of which increase the risk of preterm or LBW. Whilst some studies report either a lack of an association between stress hormone and level of cortisol secretion [117, 118], inconclusive findings [119] or increased cortisol hormone secretion at the second and third trimesters of pregnancy [120], the overall evidence suggests that hormonal and psychosocial risk factors play an important role in the cause of low with birth and stillbirth.

Consistent with other studies [76, 77, 97, 121], the risk of preterm birth was higher among women without antenatal care service during pregnancy. Adherence to antenatal care service helps for early identification and treatment of obstetric complications, severe infections and for promoting healthy behavior during pregnancy [81]. Maternal employment status also predicted the risk for preterm birth with those in paid work at a 50% higher risk compared to women performing only home duties. Logically, government or private employees would be less likely to get adequate time to attend clinical care to increase the chance of a healthy pregnancy outcome. A slightly higher proportion of employed women reported stress and fear about the delivery than non-employed participants, which was also an important risk factor for preterm birth.

Participants identifying themselves as Muslim had 1.61 times higher risk of preterm birth than those identifying as Orthodox Christian. A slightly higher proportion of those identifying as Muslim also reported receiving less support from their partner, had poor stress coping ability, fewer ANC visits, and more minor depression compared to those identifying as Orthodox Christian, and this might be a reason for Muslims to had higher preterm as these factors independently increased the risk of preterm birth.

This study presented the incidence of major adverse birth outcomes and highlighted how psychosocial risk factors play an important role in determining such outcomes. Psychological science in pregnancy [122] is an emerging research area aimed at investigating the effect of psychosocial problems on adverse birth outcomes in developed countries. Our study contributes to this research area by investigating the mediating and moderating effects of psychosocial risk factors of adverse birth outcomes in a low-income country with data from a community-based cohort study.

The study had few limitations, including the exclusion of women with a high probability of depression. In addition to reducing the overall sample of the study, this exclusion may have either underestimated or overestimated the effects of antenatal depression on the risk of adverse birth outcomes. A further limitation was the non-measurement of few possible confounders, mediators and outcomes that might have additional light on the proposed mechanisms. These variables include information on abuse, pregnancy complications (preeclampsia, gestational diabetes), history of infections, and levels of inflammation during pregnancy. However, a modified Poisson regression approach is theoretically robust to omitted covariates. The other limitation is associated with the usual challenges of measuring preterm birth and low birth weight due to the accuracy of the last menstrual cycle and weight in places other than health facilities, respectively.

Our study also had several strengths. The cohort was community-based, and the study population was created using a stratified cluster-based random sampling approach. Our results

are, therefore, likely to be representative of the incidence and factors that predict adverse pregnancy outcomes within the region. The sample size was also relatively large, although power was reduced due to the low frequency of some outcomes. Besides, we used two different approaches to assessing the associations. A regression approach allowed us to assess the overall independent effects of each risk factor, while the path analysis allowed us to assess both the direct and indirect effects of each risk factor, as well as potential mediators of the indirect effects. Finally, the use of a modified Poisson regression approach increased the precision of our estimates and allowed the reporting of relative risks rather than odds ratios, thereby increasing interpretability.

## Conclusion

The incidence of preterm birth, low birth weight, and stillbirth in Gondar Town in Ethiopia was high, suggesting the need for improved surveillance, particularly for those women with risk factors for adverse outcomes. Maternal occupation, religious identity, antenatal care service uptake, and fear of delivery were independent predictors the risk of preterm birth. Similarly, preterm birth and smoking during pregnancy both increased the risk of low birth weight. The positive association and between depression and preterm birth, and between depression and low birth weight was attenuated by partner support and increased stress coping ability. As such, interventions that focus on increasing partner engagement and participation in antenatal support, thereby enhancing maternal resilience, may reduce the risk of adverse birth outcomes in Gondar Town.

## Author Contributions

**Conceptualization:** Abel Fekadu Dadi, Emma R. Miller, Lillian Mwanri.

**Data curation:** Abel Fekadu Dadi, Richard J. Woodman, Telake Azale, Lillian Mwanri.

**Formal analysis:** Abel Fekadu Dadi, Richard J. Woodman.

**Investigation:** Emma R. Miller, Telake Azale, Lillian Mwanri.

**Methodology:** Abel Fekadu Dadi, Emma R. Miller, Telake Azale, Lillian Mwanri.

**Project administration:** Abel Fekadu Dadi.

**Software:** Abel Fekadu Dadi, Richard J. Woodman.

**Supervision:** Emma R. Miller, Telake Azale, Lillian Mwanri.

**Validation:** Abel Fekadu Dadi, Emma R. Miller, Richard J. Woodman, Telake Azale, Lillian Mwanri.

**Visualization:** Abel Fekadu Dadi, Richard J. Woodman.

**Writing – original draft:** Abel Fekadu Dadi.

**Writing – review & editing:** Abel Fekadu Dadi, Emma R. Miller, Richard J. Woodman, Telake Azale, Lillian Mwanri.

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
