## [Decision Letter · Decision Letter 0]

13 Apr 2020

PONE-D-19-29080

Effect of Antenatal Depression on Adverse Birth Outcomes in Gondar Town, Ethiopia: A community-based cohort study

PLOS ONE

Dear Dr. Abel Fekadu,

Thank you for submitting your manuscript to PLOS ONE. After careful consideration, we feel that it has merit but does not fully meet PLOS ONE’s publication criteria as it currently stands. Therefore, we invite you to submit a revised version of the manuscript that addresses the points raised during the review process.

We apologize about the delay in the review process. We invite you to resubmit as the paper received favorable reviews.

We would appreciate receiving your revised manuscript by May 28 2020 11:59PM. To enhance the reproducibility of your results, we recommend that if applicable you deposit your laboratory protocols in protocols.io, where a protocol can be assigned its own identifier (DOI) such that it can be cited independently in the future. For instructions see: http://journals.plos.org/plosone/s/submission-guidelines#loc-laboratory-protocols

We look forward to receiving your revised manuscript.

Kind regards,

Anna Palatnik, M.D.

Academic Editor

PLOS ONE

Journal Requirements:

Additional Editor Comments (if provided):

Reviewer 1:

This paper was well conceived and well-written. The authors conducted a large survey of pregnant women (n~900) in Gondar town, Ethiopia. The main exposure in the study was antenatal depression and the main outcome was incidence of adverse birth outcome (preterm birth, low-birth weight or still birth). Information on a range of risk factors (e.g. occupation, age, marital status, religion, physical activity, social support, coping strategies, antenatal healthcare attendance etc.) was also collected and including in the data analysis (which was conducted using structural equation modelling). The authors report on the main risk factors for each adverse birth outcome. The main findings in relation to the incidence of still birth indicates that Ethiopia is lagging behind other countries in reducing still births. Depression was not a risk factor for still birth in this study (perhaps due to low power); neither was is associated with low-birth weight or pre-term labour. The main findings suggest that perceived partner support and ability to cope with stress were moderated the relationship between depression and low-birth weight (although the effect was small) - and the effect was stronger when considering preterm birth. There was no main effect for experiencing depression and low-birth weight or preterm birth. So the main take-home message appears to be that interventions efforts might be better focused on increasing partner support and the woman's ability to cope with stress.

The authors neatly report on the study strengths and limitations in the discussion. The survey response rate was extremely high (97.7%).

Comments/queries:

Please provide more information on the collection of data during the second trimester. How regularly were women assessed between the second trimester and the outcome assessment (one month after birth). The data collection and the questionnaire section is somewhat confusing because it suggests that many of the study variables (e.g. partner support, stress coping ability) were assessed at the one-month post birth interview, as opposed to during the second trimester (i.e. during the pregnancy and before the birth outcome data was collected). Please clarify.

There is a bit of confusion in the discussion about the need for studies with larger sample sizes, although the large sample size of the study is listed as a study strength. It appears the authors mean that larger sample sizes are required to study rare outcomes (i.e. still birth) in future.

There appears to be a mix-up in the measurement section re the cutoff score for MUAC (18-22cm should be underweight and 22.5-31cm should be normal).

Reviewer 2:

Abstract

“The impact of psychosocial factors on pregnancy outcomes has been investigated in developed countries but few studies have been conducted in low-income countries.” This sentence does not look really important. If the authors have to report this, how can the outcome differ across culture? Why the HIC findings is not generalizable to LICs?

Introduction

The authors argued that the moderating effect of depression was not studied in Ethiopia. If this is known in other countries, is it not generalizable to other countries like Ethiopia?

Methods

The authors argued that preterm birth was not controlled in previous studies. it is understood that measurement of preterm birth is very problematic as women cannot remember the LMP which might have affected the results. The challenges related to assessment of preterm birth and low birth weight were not stated in limitations. There is also stigma related to reporting stillbirth. How could the authors managed to reduce bias related to such stigma.

The grave limitation of previous studies on the association of depression and LBW were linked to low sample size which was also the case in the current study. How different is the current study from previous studies?

Reviewers' comments:

Reviewer's Responses to Questions

**Comments to the Author**

1. Is the manuscript technically sound, and do the data support the conclusions?

Reviewer #1: Yes

Reviewer #2: Yes

2. Has the statistical analysis been performed appropriately and rigorously? 

Reviewer #1: Yes

Reviewer #2: Yes

3. Have the authors made all data underlying the findings in their manuscript fully available?

Reviewer #1: Yes

Reviewer #2: Yes

4. Is the manuscript presented in an intelligible fashion and written in standard English?

Reviewer #1: Yes

Reviewer #2: Yes

5. Review Comments to the Author

Reviewer #1: Abstract

“The impact of psychosocial factors on pregnancy outcomes has been investigated in developed countries but few studies have been conducted in low-income countries.” This sentence does not look really important. If the authors have to report this, how can the outcome differ across culture? Why the HIC findings is not generalizable to LICs?

Introduction

The authors argued that the moderating effect of depression was not studied in Ethiopia. If this is known in other countries, is it not generalizable to other countries like Ethiopia?

Methods

The authors argued that preterm birth was not controlled in previous studies. it is understood that measurement of preterm birth is very problematic as women cannot remember the LMP which might have affected the results. The challenges related to assessment of preterm birth and low birth weight were not stated in limitations. There is also stigma related to reporting stillbirth. How could the authors managed to reduce bias related to such stigma.

The grave limitation of previous studies on the association of depression and LBW were linked to low sample size which was also the case in the current study. How different is the current study from previous studies?

Reviewer #2: This paper was well conceived and well-written. The authors conducted a large survey of pregnant women (n~900) in Gondar town, Ethiopia. The main exposure in the study was antenatal depression and the main outcome was incidence of adverse birth outcome (preterm birth, low-birth weight or still birth). Information on a range of risk factors (e.g. occupation, age, marital status, religion, physical activity, social support, coping strategies, antenatal healthcare attendance etc.) was also collected and including in the data analysis (which was conducted using structural equation modelling). The authors report on the main risk factors for each adverse birth outcome. The main findings in relation to the incidence of still birth indicates that Ethiopia is lagging behind other countries in reducing still births. Depression was not a risk factor for still birth in this study (perhaps due to low power); neither was is associated with low-birth weight or pre-term labour. The main findings suggest that perceived partner support and ability to cope with stress were moderated the relationship between depression and low-birth weight (although the effect was small) - and the effect was stronger when considering preterm birth. There was no main effect for experiencing depression and low-birth weight or preterm birth. So the main take-home message appears to be that interventions efforts might be better focused on increasing partner support and the woman's ability to cope with stress.

The authors neatly report on the study strengths and limitations in the discussion. The survey response rate was extremely high (97.7%).

Comments/queries:

Please provide more information on the collection of data during the second trimester. How regularly were women assessed between the second trimester and the outcome assessment (one month after birth). The data collection and the questionnaire section is somewhat confusing because it suggests that many of the study variables (e.g. partner support, stress coping ability) were assessed at the one-month post birth interview, as opposed to during the second trimester (i.e. during the pregnancy and before the birth outcome data was collected). Please clarify.

There is a bit of confusion in the discussion about the need for studies with larger sample sizes, although the large sample size of the study is listed as a study strength. It appears the authors mean that larger sample sizes are required to study rare outcomes (i.e. still birth) in future.

There appears to be a mix-up in the measurement section re the cutoff score for MUAC (18-22cm should be underweight and 22.5-31cm should be normal).

6. PLOS authors have the option to publish the peer review history of their article (what does this mean?). If published, this will include your full peer review and any attached files.

Reviewer #1: No

Reviewer #2: Yes: Orla McBride

---

## [Author Response · Author response to Decision Letter 0]

17 Apr 2020

The response to reviewers document has been attached.

---

## [Editor Report · Decision Letter 1]

6 May 2020

PONE-D-19-29080R1

Effect of Antenatal Depression on Adverse Birth Outcomes in Gondar Town, Ethiopia: A community-based cohort study

PLOS ONE

Dear Mr Fekadu,

Thank you for submitting your manuscript to PLOS ONE. After careful consideration, we feel that it has merit but does not fully meet PLOS ONE’s publication criteria as it currently stands. Therefore, we invite you to submit a revised version of the manuscript that addresses the points raised during the review process.

Please explain how did you achieve such high response rate on the survey, Thank you

We would appreciate receiving your revised manuscript by Jun 20 2020 11:59PM. To enhance the reproducibility of your results, we recommend that if applicable you deposit your laboratory protocols in protocols.io, where a protocol can be assigned its own identifier (DOI) such that it can be cited independently in the future. For instructions see: http://journals.plos.org/plosone/s/submission-guidelines#loc-laboratory-protocols

We look forward to receiving your revised manuscript.

Kind regards,

Anna Palatnik, M.D.

Academic Editor

PLOS ONE

---

## [Author Response · Author response to Decision Letter 1]

7 May 2020

Response for editor comment has been included in journal submission portal.

---

## [Editor Report · Decision Letter 2]

2 Jun 2020

Effect of Antenatal Depression on Adverse Birth Outcomes in Gondar Town, Ethiopia: A community-based cohort study

PONE-D-19-29080R2

Dear Dr. Fekadu,

We’re pleased to inform you that your manuscript has been judged scientifically suitable for publication and will be formally accepted for publication once it meets all outstanding technical requirements.

Kind regards,

Anna Palatnik, M.D.

Academic Editor

PLOS ONE
---

## [Editor Report · Acceptance letter]

5 Jun 2020

PONE-D-19-29080R2 

Effect of Antenatal Depression on Adverse Birth Outcomes in Gondar Town, Ethiopia: A community-based cohort study 

Dear Dr. Fekadu:

I'm pleased to inform you that your manuscript has been deemed suitable for publication in PLOS ONE. Congratulations! Your manuscript is now with our production department. 

Kind regards, 

on behalf of

Dr. Anna Palatnik 

Academic Editor

PLOS ONE